# Exosomal let-7e, miR-21-5p, miR-145, miR-146a and miR-155 in Predicting Antidepressants Response in Patients with Major Depressive Disorder

**DOI:** 10.3390/biomedicines9101428

**Published:** 2021-10-09

**Authors:** Yi-Yung Hung, Chen-Kai Chou, Yi-Chien Yang, Hung-Chun Fu, El-Wui Loh, Hong-Yo Kang

**Affiliations:** 1Department of Psychiatry, Kaohsiung Chang Gung Memorial Hospital, College of Medicine, Chang Gung University, Kaohsiung 833, Taiwan; 2Division of Endocrinology and Metabolism, Department of Internal Medicine, Kaohsiung Chang Gung Memorial Hospital, College of Medicine, Chang Gung University, Kaohsiung 833, Taiwan; chou@cgmh.org.tw; 3Graduate Institute of Clinical Medical Sciences, Chang Gung University, Kaohsiung 833, Taiwan; 4Department of Dermatology, Kaohsiung Chang Gung Memorial Hospital, College of Medicine, Chang Gung University, Kaohsiung 833, Taiwan; yichienyang@gmail.com; 5Department of Obstetrics and Gynecology, Kaohsiung Chang Gung Memorial Hospital, College of Medicine, Chang Gung University, Kaohsiung 833, Taiwan; allen133@cgmh.org.tw; 6Center for Evidence-Based Health Care, Department of Medical Research, Taipei Medical University Shuang Ho Hospital, New Taipei City 235, Taiwan; lohew@hotmail.com; 7Department of Dentistry, Taipei Medical University Shuang Ho Hospital, New Taipei City 235, Taiwan; 8Cochrane Taiwan, Taipei Medical University, Taipei 110, Taiwan; 9Graduate Institute of Clinical Medicine, College of Medicine, Taipei Medical University, Taipei 110, Taiwan

**Keywords:** exosome, microRNA, major depressive disorder

## Abstract

The intracellular microRNAs that negatively regulate Toll-like receptor 4 signaling pathways in peripheral blood mononuclear cells are associated with major depressive disorder (MDD). However, that the distribution of these microRNAs in exosomes could be a biomarker of central nervous system diseases is just beginning to be explored. In the present study, we isolated serum exosomes from patients with MDD and healthy controls to explore the levels of exosomal microRNAs, including let-7e, miR-21-5p, miR-223, miR-145, miR-146a, and miR-155. We also investigated the changes of these exosomal microRNAs after antidepressant treatment and their association with clinical changes in scores on the Hamilton Depression Rating Scale. An ANCOVA adjusted by age, sex, BMI, and smoking showed higher expression levels of miR-146a (*p* = 0.006) in patients with MDD compared to controls. Patients who achieved remission showed significantly lower let-7e, miR-21-5p, miR-145, miR-146a, and miR-155 levels before treatment and increased levels after antidepressant treatment compared with the non-remission group. Through receiver operating characteristic (ROC) analysis, let-7e, miR-145, and miR-146a showed acceptable discrimination between the remission and non-remission groups, whereas miR-21-5p and miR-155 showed poor discrimination. These findings demonstrate that exosomal microRNAs may play essential roles in predicting antidepressants response.

## 1. Introduction

Depression is among the most common and costly of all psychiatric disorders. Nearly one in four women and one in six men experience at least one clinically defined depression episode in their lifetimes. Despite decades of research effort, there are still no biomarkers that can be used to monitor treatment responses, disease states, or predict individual responses to specific drugs [1].

Major depression is etiologically related to the peripheral and central innate immune systems, especially aberrant Toll-like receptor (TLR) expression [2,3,4]. In addition, the negative regulators of TLR signal pathways that prevent TLR-induced cytokine storms by controlling the magnitude and duration of responses play essential roles in the development of major depressive disorder (MDD) [5,6]. Exosomes are thought to be able to change the behavior of recipient cells and act as key players in many cellular and molecular pathways in mental disorders [7,8]. Plasma exosomes can deliver sigma-1 receptors to the central nervous system, ameliorate inflammation, and improve depressive-like behavior in animals [9]. The differential expression of microRNAs in the exosomes of the rat model was associated with the MAPK pathway, Wnt pathway, and mTOR pathway [10]. Alexander et al. (2015) found that intracellular miR-155 and miR-146a are released into exosomes and are subsequently taken up by target cells. Following uptake, microRNAs regulate the cellular response to lipopolysaccharide, which activates the TLR4 signaling pathways that exosome-delivered miR-155 enhances, while miR-146a reduces inflammatory gene expression [11].

In our previous work, we measured the intracellular microRNAs that regulate TLR4 signaling. We found that let-7e, miR-146a, and miR-155 were lower in MDD patients than in healthy controls and were significantly higher after than before antidepressants treatment [6]. In the same study, we noticed that let-7e and miR-146a were negatively correlated with one another, but miR-155 expression was positively correlated with the severity of depression. In the current study, we investigated the microRNA expression profiles of negatively regulating TLR4 signaling, including the measurement of let-7e, miR-21-5p, miR-223, miR-145, miR-146a, and miR-155 in the serum exosome obtained from patients with MDD before and after treatment with antidepressants.

## 2. Materials and Methods

### 2.1. Participants

The Institutional Review Board and hospital ethics committee of Kaohsiung Chang Gung Memorial Hospital approved the current study protocol (201700539A3, 16 March 2017 and 201901894A3, 21 January 2020). Hospitalized in patients with MDD from the psychiatric ward of Kaohsiung Chang Gung Memorial Hospital, Taiwan, were enrolled from August 2013 to August 2018. The patients and healthy controls agreeing to join the study provided written informed consent. Serum samples before and after antidepressant treatments were obtained from MDD patients, and single serum samples were obtained from the healthy control subjects. Patients were hospitalized in the psychiatric ward of Kaohsiung Chang Gung Memorial Hospital and monitored for drug adherence, strict regular sleep–wake cycles, a well-controlled diet, and limited smoking.

Participants were aged 20–65 years and were medically healthy based on their clinical history, physical examination, negative routine blood tests, and urine examination. MDD patients were screened with the structured clinical interview for the Diagnostic and Statistical Manual of Mental Disorders, Fifth Edition (DSM-V) Axis I Disorders by two psychiatrists before entering the study. The severity of MDD before and after antidepressant treatments was assessed by the same psychiatrists using the 17-item Hamilton Depression Rating Scale (HAMD-17). Remission was defined as a total HAMD-17 score ≤ 7. Patients with limited capacity, psychotic disorder, mental retardation, substance dependence, a body mass index (BMI) >34 kg/m^2^, a history of any systemic inflammatory disease, or a history of taking anti-inflammatory or immune-modulating drugs were excluded from the study. No antidepressants were taken for at least one week before they entered the study. The healthy control subjects were screened by these two psychiatrists using the Diagnostic and Statistical Manual of Mental Disorders (Fifth Edition). Those who had have a personal or family history (first-degree relative) of psychiatric disorder were excluded.

### 2.2. Treatment

Choice of antidepressants was based on clinical considerations and was not influenced by the study. Chosen antidepressants were administered and recorded after screening at baseline. The chosen antidepressants were escitalopram (10–20 mg/day; *n* = 5), fluoxetine (40–80 mg/day; *n* = 3), paroxetine (20–40 mg/day; *n* = 7), sertraline (75–140 mg/day; *n* = 3), duloxetine (60–90 mg/day; *n* = 5), venlafaxine (37.5–225 mg/day; *n* = 4), bupropion (150–300 mg/day; *n* = 6), mirtazapine (30–45 mg/day; *n* = 5), and agomelatine (50 mg/day; *n* = 1). Benzodiazepines were restricted from use using the research period. In addition, patients were provided with one or two supportive psychotherapy sessions during the study and urged to participate in regular activities while hospitalized.

### 2.3. Serum Exosomes Isolation

An ExoQuick exosome isolation kit (System Biosciences, Cat#RA806A-1, Mountain View, CA, USA) was used to isolate the serum exosome RNA per the manufacturer’s instruction. Briefly, 500 μL serum was combined with 120 μL of ExoQuick solution and placed at 4 °C for 30 min. The mixture was centrifuged at 13,000 rpm for 2 min. A 350 μL LYSIS buffer was added to the exosome pellet and vortexed for 15 s. Then, 100% ethanol was added, and the entire volume of the solution was transferred to the spin column. After two washes, the exosome RNA was eluted with the ELUTION buffer for further analysis.

### 2.4. Exosomal RNA Extraction

Total RNA was extracted from the serum exosomes using the SeraMir Exosome RNA Amplification kit (System Biosciences, Cat#RA806A-1) according to the manufacturer′s protocol, and reverse-transcribed to complementary DNA (cDNA) using the TaqMan Micro RNA Transcription Kits (Applied Biosystems 4366596, Foster City, CA, USA). The thermal cycling conditions are 16 °C for 30 min, 42 °C for 30 min, and 85 °C for 5 min.

#### 2.4.1. Quantitative Reverse-Transcription Polymerase Chain Reaction (qRT-PCR)

Specific sequences (Table 1) were amplified in an Applied Biosystems 7500 system using the TaqMan^®^ Universal Master Mix II, without UNG (Applied Biosystems 4440040). The amplification conditions included an initial denaturation at 95 °C for 10 min, followed by 40 cycles of denaturation at 95 °C for 15 s, and annealing and extension at 60 °C for 1 min. Gene expression was quantified using the software provided by the manufacturer (Applied Biosystems), analyzed according to the 44 Ct method, and normalized to the expression of U6 small nuclear RNA (snRNA).

#### 2.4.2. Statistical Analysis

All results are presented as mean ± standard deviation (SD). Statistical analyses were performed using Statistical Product and Service Solutions (SPSS) version 22. Data, including age, body mass index (BMI), and microRNAs in depression and health control groups, were analyzed with a Shapiro–Wilk test, which revealed a normal distribution. Age and BMI were compared using Student’s *t*-tests. Sex and smoking were compared using the chi-squared tests. Between-group differences were assessed using the analysis of covariance (ANCOVA) following adjustment for age, sex, smoking, and BMI. The effects of antidepressants on microRNA levels before and after treatments were tested using a paired *t*-test. A Mann–Whitney U test was used to compare the difference between the subgroups (remission and non-remission) and health control. A Wilcoxon signed-rank test was used to compare the effects of antidepressants in the remission and non-remission groups. Furthermore, the area under the curve (AUC) value, obtained by performing the receiver operating curve (ROC) analysis, was analyzed to see whether the microRNAs may predict the non-remission of the patients.

## 3. Results

### 3.1. Demographics

A total of 52 patients with MDD, including 18 men and 34 women, were enrolled. Of these, 39 patients were treated with an antidepressant for four weeks and received a follow-up examination. Age and BMI were similar in healthy controls and MDD patients before treatment. The smoking rate was significantly higher in the MDD group and the HAMD-17 scores after antidepressants treatment (7.53 ± 4.27) were significantly lower before the antidepressant treatment (23.35 ± 4.76) (Table 2). When patients were stratified by remission or non-remission, there was no difference in age, sex, BMI, smoking, and HAMD score before treatment.

### 3.2. Different Expression Profiles of Exosomal microRNA between Remission and Non-Remission

Our previous work found that intracellular microRNAs, including let-7e, miR-223, miR-145, and miR-155, are associated with remission. Since exosomes can encapsulate microRNA to prevent enzyme degradation during transference and then release them into the recipient cells [12], we hypothesized that intracellular changes could affect exosomal microRNA expression. Through an ANCOVA adjusted by age, sex, BMI, and smoking, we initially found higher expression levels of miR-146a (*p* = 0.006) in patients with MDD compared with controls (Table 3).

To further investigate the association between the changes of exosomal microRNA and responses to antidepressants treatment, we divided the patients into remission and non-remission groups. Patients who achieved remission showed significantly increased levels of let-7e, miR-21-5p, miR-223, miR-145, miR-146a, and miR-155 after treatment (Table 3, Figure 1) but lower levels of let-7e, miR-21-5p, miR-145, miR-146a, and miR-155 before treatment (Table 4, Figure 1).

Multiple lines of evidence have indicated that exosomal microRNAs act as key players in many cellular and molecular pathways involved with mental disorders [7]. In addition, some microRNAs were thought to be biomarkers for MDD [13]. To understand the predictability of exosomal microRNAs on antidepressant drug responses, we conducted a ROC analysis (Figure 2) to estimate the sensitivity and specificity of these microRNAs. As shown in Figure 2, the ROC curves of let-7e (AUC = 0.789, 95% confidence interval (CI) = 0.643–0.935), miR-145 (AUC = 0.711, 95% confidence interval (CI) = 0.547–0.875), and miR-146a (AUC = 0.759, 95% confidence interval (CI) = 0.607–0.910) reflected acceptable discrimination between remission and non-remission, whereas miR-21-5p and miR-155 showed poor discrimination.

## 4. Discussion

This study assessed the associations between MDD and the expression of exosomal microRNAs that regulate TLR4 signaling. We found that, prior to antidepressant treatment, only miR-146a was significantly lower in MDD patients than in healthy controls. In addition, let-7e and miR-155 were increased by treatment with antidepressants. However, when we further divided the patients into remission and non-remission groups, we noticed that the expression profile of the remission group was similar to that of the control group at baseline except for miR-155, which showed significantly lower levels in the remission group. All exosomal microRNA increased significantly after antidepressant treatment. In contrast to the remission group, the non-remission group showed significantly higher let-7e, miR-21-5p, miR-223, and miR-146a than healthy controls. This study is the first to explore exosomal negative regulatory microRNAs for TLR4 signaling and show increases after antidepressant treatment.

Regarding the role of microRNAs as predictors for antidepressant responses, our previous work found no difference in intracellular microRNAs between the remission and non-remission groups at baseline [6]. However, In the current work, the results of exosomal microRNAs, including let-7e, miR-125a, miR-145, miR-146a, and miR-155 levels, showed predictability in achieving remission. Kim et al. (2019) also identified a rise of baseline plasma miR-146a-5p and miR-21-5p levels in MDD patients who entered remission with duloxetine treatment [14].

The effects of antidepressants on miR-155 was reported by J Dai (2020), who showed that fluoxetine treatments significantly reduce miR-155 expression in the hippocampus in mice [15]. Xun Wang (2018) also reported that citalopram downregulates serum miR-155. However, in this work, we noticed the similar findings to our previous work, which showed that miR-155 is increased only in the remission group after the administration of antidepressants [6]. The source of the sample is one reason for this difference.

The effects of antidepressants on miR-146a-5p differed in different studies with different sources. Downregulation of total miR-146a-5p was observed among escitalopram responders compared to non-responders [16]. Marshe et al. (2020) reported that venlafaxine did not change intracellular miR-146a expression. However, our study, which investigates exosomal miR-146a-5p, found that patients who achieved remission showed a significant increase after antidepressant treatments.

The microRNAs in cell-released exosomes can circulate to reach neighboring cells and distant target cells [17]. The exosomes miR-146a and let-7e downregulate inflammatory signals [18,19]. Our study revealed higher exosomal let-7e and miR-146a in non-remission patients, and antidepressant treatments could not enhance the mRNA levels. These suggest that antidepressants may ameliorate inflammation via negative regulatory microRNA. Small symptom changes might have been unseen through human-based observations because the target microRNA is already upregulated, and this creates a ceiling effect. The function of miR-155 is to promote inflammation [20]. In our study, miR-155 was not downregulated in the acute stage of depression, suggesting that the spontaneous balance function in the non-remission group is impaired. Much evidence has shown that microRNAs play important roles in modulating drug metabolizing enzymes and transporters, which causes interindividual variability. Baseline biomarkers of miRNA levels are not only useful in diagnosing MDD, but also help predict the drug responses. Some studies have reported a relationship between microRNAs and drug responsiveness in MDD. For instance, patients who responded to treatment with citalopram had lower levels of miR-1202 at baseline than the non-responders, while the expression of miR-1202 increased during the course of the treatment in responders [21]. One study showed that 30 miRNAs were differentially expressed after 12 weeks of escitalopram treatment in the whole blood sample of 10 subjects with depression [22]. It has also been shown that miRNA-451a miRNA-34a-5p, and miRNA-221-3p levels are related to paroxetine efficacy [23]. Exosomal miR-139-5p levels discriminated between MDD and healthy subjects with a sensitivity of 0.867 and specificity of 0.767, and an AUC of 0.807 [24].

The main limitation of our study is that the origins and the targets of the serum exosome we collected are unknown. The exosomes in serum that we detected are the total exosomes derived from different systems, including the central nervous system and peripheral blood. Different targeting of these exosomes may cause different results which could lead to different interpretations compared to our work. Second, we did not examine the intracellular, serum, and exosome levels of microRNAs in the same sample at the same time. Thus, there might be a bias in correlation between the indicators. Third, different types of antidepressants were used by the patients, and this may confound the results. In addition, the number of patients is relatively small and further investigation with a larger number of patients is needed to confirm these findings.

## 5. Conclusions

This study is an extension of our previous work regarding negative regulatory intracellular microRNA. Our results not only showed that exosomal negative regulatory microRNAs are good parameters for antidepressants treatment, but that they also demonstrate acceptable performance in predicting remission and non-remission in patients prior to antidepressant treatment. Further studies that include a larger sample size with different category groups of doses and types of antidepressants, with a focus on the effects of exosomes in innate immune regulation, are warranted.

## Figures and Tables

**Figure 1 biomedicines-09-01428-f001:**
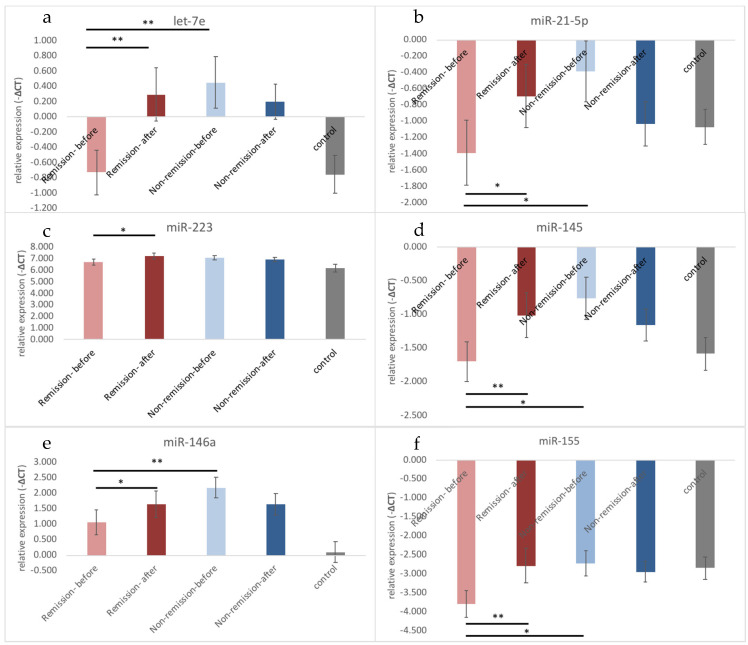
The difference of microRNA between remission and non-remission in patients with major depressive disorder: (**a**) let-7e; (**b**) miR-21-5p; (**c**) miR-223 (**d**) miR-145; (**e**) miR-146a; (**f**) miR-155. Before antidepressants treatment, let-7e, miR-21-5p, miR-145, and miR-146a were significantly lower than non-remission group but were not different from health controls. Antidepressants significantly increased let-7e, miR-21-5p, miR-223, miR-145, miR-146a and miR-155 expression levels.* *p* < 0.05; ** *p* < 0.01.

**Figure 2 biomedicines-09-01428-f002:**
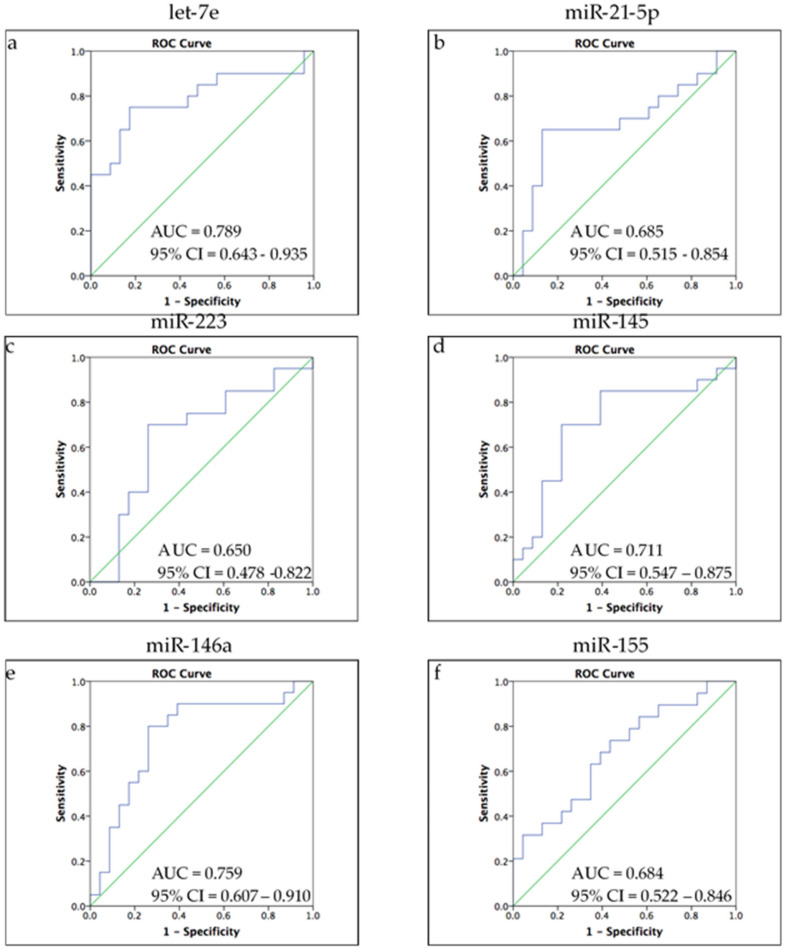
ROC analysis of ability of microRNAs to discriminate remission from non-remission. (**a**) let-7e; (**b**) miR-21-5p; (**c**) miR-223 (**d**) miR-145; (**e**) miR-146a; (**f**) miR-155. (**a,d,e**) let-7e, miR-145, and miR-146a showed acceptable discrimination (0.7 ≥ AUC > 0.6) between remission and non-remission (**b,f**) miR-21-5p and miR-155 showed poor discrimination (0.6 ≥ AUC > 0.5). (**c**) miR-223 showed no discrimination.

**Table 1 biomedicines-09-01428-t001:** Primers for qRT-PCR.

Assay ID	Assay Name	Mature microRNA Sequence
002406	hsa-let-7e	UGAGGUAGGAGGUUGUAUAGUU
000397	hsa-miR-21-5P	UGAGGUAGUAGGUUGUAUGGUU
002198	hsa-miR-125a-5p	UCCCUGAGACCCUUUAACCUGUGA
002278	hsa-miR-145	GUCCAGUUUUCCCAGGAAUCCCU
000468	hsa-miR-146a	UGAGAACUGAAUUCCAUGGGUU
002623	hsa-miR-155	UUAAUGCUAAUCGUGAUAGGGGU
002285	hsa-miR-186	CAAAGAAUUCUCCUUUUGGGCU
002295	hsa-miR-223	UGUCAGUUUGUCAAAUACCCCA
001973	U6 snRNA	GTGCTCGCTTCGGCAGCACATATACTAAAATTGGAACGATACAGAGAAGATTAGCATGGCCCCTGCGCAAGGATGACACGCAAATTCGTGAAGCGTTCCATATTTT

**Table 2 biomedicines-09-01428-t002:** Demographic and clinical characteristics of patients with major depressive disorder and health control.

	Depression before Treatment (*n* = 52)	Health Control (*n* = 31)	*p*-Value
Age (years)	42.52 ± 13.51	39.06 ± 7.95	0.199
Sex (M/F)	18/34	9/22	0.637
BMI (kg/m^2^)	24.57 ± 4.56	23.24 ± 2.61	0.142
Smoking (yes/no)	20/32	2/29	0.002 **
HAMD before treatment	23.35 ± 4.76	-	-
HAMD after treatment	7.53 ± 4.27	-	-

Results reported as a mean ± SD or as a number; age and BMI were compared using Student’s *t* tests; sex and smoking were compared by the chi-square test; ** *p* < 0.01.

**Table 3 biomedicines-09-01428-t003:** The difference of serum exosomal microRNA expression between health control and major depression groups before and after treatment.

	I	II	III	I vs III	I vs II
	Depression before treatment *n* = 52	Depression after treatment *n* = 39	Health control *n* = 31	*p*-value	*p*-value
let-7e	−0.243 ± 1.466	0.258 ± 1.268	−0.752 ± 1.351	0.443	0.044 *
Remission	−0.727 ± 1.160	0.295 ± 1.390		0.900	0.002 ^§§^
Non-remission	0.452 ± 1.611	0.205 ± 1.112		0.009 ^¶,¶^	0.278
miR-21-5p	−0.977 ± 1.730	−0.823 ± 1.459	−1.07 ± 1.18	0.605	0.581
Remission	−1.390 ± 1.594	−0.690 ± 1.571		0.396	0.036 ^§^
Non-remission	−0.384 ± 1.794	−1.036 ± 1.302		0.122	0.098
miR-223	6.86 ± 1.019	7.140 ± 0.964	6.19 ± 1.68	0.127	0.102
Remission	6.704 ± 1.081	7.254 ± 1.025		0.416	0.014 ^§^
Non-remission	7.084 ± 0.910	6.976 ± 0.874		0.007 ^¶,¶^	0.278
miR-145	−1.316 ± 1.394	−1.068 ± 1.230	−1.58 ± 1.32	0.765	0.274
Remission	−1.700 ± 1.190	−1.009 ± 1.308		0.693	0.003 ^§§^
Non-remission	−0.763 ± 1.514	−1.154 ± 1.145		0.149	0.179
miR-146a	1.515 ± 1.685	1.639 ± 1.681	0.10 ± 1.83	0.007 *	0.592
Remission	1.051 ± 1.627	1.642 ± 1.756		0.062	0.048 ^§^
Non-remission	2.181 ± 1.583	1.635 ± 1.624		0.000 ^¶¶^	0.079
miR-155	−3.353 ± 1.554	−2.849 ± 1.627	−2.85 ± 1.67	0.126	0.048 *
Remission	−3.794 ± 1.408	−2.780 ± 1.840		0.081	0.004 ^§§^
Non-remission	−2.719 ± 1.576	−2.949 ± 1.311		0.995	0.408

*,**: ANCOVA adjust with age, sex, BMI, and smoking; ^¶^, ^¶¶^: Mann–Whitney U test; ^§^, ^§§^: Wilcoxon signed-rank test; *, ^§^ *p* < 0.05; **, ^§§^, ^¶¶^ *p* < 0.01.

**Table 4 biomedicines-09-01428-t004:** Difference of exosomal microRNA expression between remission and non-remission at baseline.

	Remission (*n* = 16)	Non-Remission (*n* = 23)	*p*-Value
let-7e	−0.727 ± 1.160	0.452 ± 1.611	0.001 **
miR-21-5p	−1.390 ± 1.594	−0.384 ± 1.794	0.038 *
miR-125a	−4.706 ± 1.411	−3.545 ± 1.493	0.018 *
miR-223	6.704 ± 1.081	7.084 ± 0.910	0.093
miR-145	−1.700 ± 1.190	−0.763 ± 1.514	0.018 *
miR-146a	1.051 ± 1.627	2.181 ± 1.583	0.004 **
miR-155	−3.794 ± 1.408	−2.719 ± 1.576	0.042 *

Mann–Whitney U test; * *p* < 0.05; ** *p* < 0.01.

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
