# Peer review of "Exosomal let-7e, miR-21-5p, miR-145, miR-146a and miR-155 in Predicting Antidepressants Response in Patients with Major Depressive Disorder"

_biomedicines, 2021, doi:10.3390/biomedicines9101428_

Round 1
Reviewer 1 Report
ADGRG1/GPR56 (a G protein coupled receptors GPCRs) have been implicated in diverse pathological processes and tumor progression.
Authors discussed it’s detail role and covered latest discoveries in the field, specially focusing on the tumor biology. The current review is written nicely with minimal figures, but similar article by the same group overlap the presented data ( https://doi.org/10.1016/j.bj.2021.04.012 ). Although these two papers discuss separate biology of GPCRs, but it’s important to make clear statement on these articles about any overlap results and discussion.
Once authors satisfy this overlap finding, it should be great addition to current understanding on GPCRs in tumor and other health related issues.
Author Response
Thanks for reviewing our manuscript. However, our manuscript is associated with microRNA expression in exosome from patients with major depressive disorder rather than G protein coupled receptors (GPCRs). Therefore, I can not response to suggestions appropriatly.
Reviewer 2 Report
The authors present novel insight regarding miR and MDD. While the manuscript seems interesting, some aspects need to be addressed before accepting the paper for publication.
-Figure 1 is barely readable.
-did the authors check for normal distribution when using the T-test? if so, a general statement should be added to the material section.
The authors are especially to be gratulated, as they correctly delineate the limitations of their work and they do not try to overclaim their findings.
Author Response
Response to the reviewers’ comments
The referees gave our paper positive comments by mentioning “The authors present novel insight regarding miR and MDD. While the manuscript seems interesting, some aspects need to be addressed before accepting the paper for publication.” Here are our responses to improve the manuscript, in light of the reviewer’s comments. (Point-by-point).
Point 1: -Figure 1 is barely readable.
Response 1: Thanks for your suggestion. We already improved the resolution of figure 1 and figure 2.
Point 2: -did the authors check for normal distribution when using the T-test? if so, a general statement should be added to the material section.
Response 2: Thanks for your suggestion. Normal distribution were checked before using the T-test. Therefore, we added a paragraph to address this point. “Age and body mass index (BMI) were compared using the Student’s t-tests after checking normal distribution.” (page 4, line 139-141) .